# Effects of Ionizing Radiation on Cardiac Implantable Electronic Devices (CIEDs) in Patients with Esophageal Cancer Undergoing Radiotherapy: A Pilot Study

**DOI:** 10.3390/cancers16030555

**Published:** 2024-01-28

**Authors:** Davut D. Uzun, Janek Salatzki, Panagiotis Xynogalos, Norbert Frey, Juergen Debus, Kristin Lang

**Affiliations:** 1Department of Anesthesiology, Heidelberg University Hospital, 69120 Heidelberg, Germany; deniz.uzun@med.uni-heidelberg.de; 2Heidelberg Center for Heart Rhythm Disorders (HCR), 69120 Heidelberg, Germany; janek.salatzki@med.uni-heidelberg.de (J.S.); panagiotis.xynogalos@med.uni-heidelberg.de (P.X.); norbert.frey@med.uni-heidelberg.de (N.F.); 3Department of Cardiology, Heidelberg University Hospital, 69120 Heidelberg, Germany; 4Department of Radiation Oncology, Heidelberg University Hospital, 69120 Heidelberg, Germany; juergen.debus@med.uni-heidelberg.de; 5Heidelberg Institute of Radiation Oncology (HIRO), 69120 Heidelberg, Germany; 6National Center for Tumor Diseases (NCT), 69120 Heidelberg, Germany; 7Department of Radiation Oncology, Heidelberg Ion-Beam Therapy Center (HIT), Heidelberg University Hospital, 69120 Heidelberg, Germany; 8Clinical Cooperation Unit Radiation Oncology, German Cancer Research Center (DKFZ), 69120 Heidelberg, Germany; 9German Cancer Consortium (DKTK), Partner Site Heidelberg, 69120 Heidelberg, Germany

**Keywords:** radiotherapy, esophageal cancer, cardiac devices, toxicity

## Abstract

**Simple Summary:**

The number of cancer patients who require an implantable electronic heart device due to their heart disease is steadily increasing. Radiotherapy is an important treatment option for esophageal cancer, in addition to surgery and chemotherapy. Ionizing radiation can lead to serious malfunctions of cardiac devices, including complete loss of function with life-threatening consequences. There is already data in the literature on other types of cancer and radiotherapy. Esophageal cancer is rare, so there is limited information on patients with this cancer who need radiotherapy and have an electronic heart device. We retrospectively analyzed the occurrence of cardiac device malfunction in patients with esophageal cancer in our study between 2012 and 2022. By applying our internal treatment protocol, serious cardiac device malfunctions during, and after radiotherapy could be prevented. It is important to carefully plan and select the energy of the photons. The manufacturer’s recommended dose limits should be observed.

**Abstract:**

(1) Background: The prevalence of cancer patients relying on cardiac implantable electronic device (CIED) is steadily rising. The aim of this study was to evaluate RT-related malfunctions of CIEDs. (2) Methods: We retrospectively analyze sixteen patients with esophageal cancer who were treated with radiotherapy between 2012 and 2022 at the University Hospital Heidelberg. All patients underwent systemic evaluation including pre-therapeutic cardiological examinations of the CIED functionality and after every single irradiation. (3) Results: Sixteen patients, predominantly male (14) with a mean age of 77 (range: 56–85) years were enrolled. All patients received 28 fractions of radiotherapy with a cumulative total dose 58.8 Gy. The mean maximum dose at the CIEDs was 1.8 Gy. Following radiotherapy and during the one-year post-radiation follow-up period, there were no registered events associated with the treatment in this evaluation. (4) Conclusion: The study did not observe any severe CIED malfunctions following each radiation fraction or after completion of RT. Strict selection of photon energy and alignment with manufacturer-recommended dose limits appear to be important. Our study showed no major differences in the measured values of the pacing threshold, sensing threshold and lead impedance after RT.

## 1. Introduction

The prevalence of cancer patients relying on a cardiac implantable electronic device (CIED) is steadily rising, a trend related to changing demographics within the population [1,2,3]. With an increase in new cancer cases and the simultaneous reduction in cancer-related mortality due to advancements in therapy and early detection strategies, it is anticipated that the population of individuals living with cancer and those surviving in the long term will increase in the future. The prevalence of cardiac diseases has increased due to the rise in life expectancy, resulting in a higher number of patients with CIEDs. In fact, over 700,000 operations for newly implanted pacemakers (PMs) and implantable cardioverter defibrillators (ICDs) are performed annually worldwide. In Germany alone, approximately 150,000 CIEDs were implanted [4]. Technical advancements in these devices have extended the life expectancy of patients with previously fatal arrhythmias. The incidence and prevalence of malignant diseases are increasing in patients with cardiac implantable electronic devices (CIEDs) due to overlapping risk factors with heart disease, such as smoking, an unhealthy diet, a physically passive lifestyle, diabetes, obesity, and increasing age [5]. These risk factors increase the likelihood of developing both cancer and heart disease [6]. It is important to note that this statement is based on objective evidence and not subjective evaluation. There is mounting evidence that cardiovascular diseases and cancer share similar biological mechanisms of development [6,7]. Furthermore, cancer has been shown to increase mortality rates from heart failure [7]. In the past century, squamous cell carcinoma (SCC) was the most common type of cancer found in the esophagus and gastro-esophageal junction. However, in industrialized countries, the incidence of adenocarcinoma (AC) is increasing [8]. This shift in risk factors is likely responsible for the increase in AC cases [9]. As over two-thirds of patients are diagnosed with locally advanced or metastasized disease, the 5-year overall survival rate is relatively low, ranging from 15% to 25% [10]. In addition to surgery and chemotherapy, radiotherapy (RT) is an important option in the therapy of esophageal cancer [11,12,13]. The treatment of esophageal cancer depends heavily on the clinical stage of the tumor and regularly requires a multidisciplinary assessment. Neoadjuvant chemoradiotherapy improves overall survival in patients with respectable locally advanced esophageal cancer compared with surgery alone [12,14,15]. Ionizing radiation can lead to CIED malfunction up to a complete loss of function with life-threatening consequences [16,17]. Therefore, patients with CIED should be consulted by a cardiologist/electrophysiologist prior to first radiotherapy fraction, during radiotherapy and at the end of treatment [18]. Modern therapy options such as intensity modulated radiotherapy (IMRT) and volume modulated radiotherapy (VMAT) allow for more conformal target volumes, thereby minimizing radiation exposure to organs at risk and adjacent structures. From the data in the literature, 6 MV IMRT or VMAT plans can be applied equally well to deep tumors, such as the esophagus, with good dose distribution compared to 3D-CRT. Complicated shielding maneuvers or even relocation of CIEDs is often no longer necessary here.

The indication for pacemaker implantation (PM) is usually persistent bradycardic arrhythmias or asystolia [16]. When considering the risk of CIED failure, patient characteristics, particularly their PM dependence, must be considered. In pacemaker-dependent patients, loss of pacing can cause symptomatic bradycardia. On the other hand, loss of pacing control may lead to ventricular tachycardia.

For patients with heart failure with reduced left ventricular ejection fraction (HFrEF) and prolonged QRS time, cardiac resynchronization devices (CRT) are often recommended [16,19]. Implantable cardioverter defibrillators (ICDs) are recommended in patients who suffer from a permanent risk of malignant tachycardic arrhythmias associated with a risk of sudden cardiac death or survived sudden cardiac death [20]. Compared to PM, ICDs appear to be more susceptible to malfunction in the context of radiation therapy [4]. The causes of malfunction of CIEDs in the context of RT are most likely due to effects from direct or scattered ionizing radiation. Electromagnetic disturbance poses minimal risks to current CIEDs due to advancements in technology, increased device reliability and reduced energy consumption. However, some studies revealed an increase in the radiation sensitivity of these modern devices compared with older CIEDs independently of type of either direct or scattered ionizing radiation [21,22]. The range of potential CIED malfunctions induced by RT can vary from issues that manifest solely during radiation exposure to instances of reverting to backup mode and permanent malfunctions. The spatial proximity of the irradiation region to the CIED represents a relevant risk factor of possible malfunction remains a significant consideration. Yet, the literature concerning CIEDs and their interaction with radiotherapy in esophageal cancer is limited, leaving gaps in our understanding of how this treatment affects CIED functionality. The objective of this study is to investigate RT-related dysfunctions of CIEDs in patients with esophageal cancer

## 2. Materials and Methods

### 2.1. Evaluation

This retrospective study was performed following institutional guidelines and the Declaration of Helsinki of 1975 in its most recent version. Ethical approval for the study was granted by the local ethics committee of the medical faculty of Heidelberg University Hospital on 22 March 2021 and is listed there under the file number (S-123/2021). Patient confidentiality has been maintained by anonymizing patient data to remove any identifying information. Thus, patient consent was not required. Clinical, operative, and hospital course records were reviewed. A local databank was screened for patients with CIED, esophageal cancer, and RT between 2012 and 2022 at the University Hospital Heidelberg. RT was performed at the Department of Radiation Oncology at University Hospital Heidelberg using either intensity-modulated radiotherapy (IMRT) or three-dimensional radiotherapy (3D-CRT).

### 2.2. CIED Details

Prior to RT, all patients underwent systemic evaluation of the CIED functionality at the Department of Cardiac Electrophysiology of the Heidelberg University Hospital. Regarding CIED information, data collection at baseline (pre-RT) was performed including for different types of CIEDs (PM, ICD, CRT). Also, device location, manufacturer, number of leads, primary indication for implantation, PM dependency (defined as intrinsic rhythm ≤ 30 bpm), and history of ventricular arrhythmias (for ICD) were recorded. Naturally, as with any technical measurement, there are also uncertainties in the CIED measurements. There are no universally valid measurement uncertainties available here, rather it depends on the individual CIED and its measuring devices. After consultation with the manufacturers, we were able to provide the following measurement uncertainties for the most common CIED used in our study: Pacing impedances +/−10% with resolution of 19 ohm, shock impedances +/−10% with resolution of 2 ohms. Threshold +/−0.5 V relative to manual threshold for thresholds less than 2.5 V and +/−1.0 V for higher thresholds.

### 2.3. Treatment Planning and Treatment Characteristics

During RT-treatment planning process, all patients underwent CT simulation. RT was administered using 6 MV photons after treatment planning process, no neutron damage was possible at these energy levels. There was no patient treated with electrons. Clinical target volume definition was based on contrast-enhanced CT scans, included the primary tumor region as well as nodal involvement according to the International Commission on Radiation Units and Measurements (ICRU) definition for esophageal cancer [23]. A planning target volume (PTV) margin of 5 to 8 mm was added depending on the applied technique. In the planning CT the CIED was encompassed for all patients completely. In planning process, all CIEDs were restricted to total doses of 3.0 Gy or less and were not located within the treatment field. Relocation of the device is generally considered when this precondition is not possible, which was not the case in our study cohort. In the entire study group, analyses were performed to evaluate the distance of the RT plan and the dose at the CIED (D95). An example of a patient with esophageal carcinoma located very close to the CIED and cervical lymph node metastases is shown in Figure 1 to illustrate that in radiation planning, distances from the tumor and CIED are important for dosimetric planning. There was no need to use asynchronous stimulation of pacemakers or deactivation of antitachycardia treatment of ICD’s through reprogramming or magnet placement. We did not perform continuous monitoring by electrocardiography.

### 2.4. CIED Follow-Up

All patients were seen by a specialist at the department of Cardiac Electrophysiology of the Heidelberg University Hospital after every single fraction and 6 weeks as well as 12 months after Radiotherapy. The center’s heart specialists have a standard examination procedure for all patients with CIED after radiotherapy. This ensures objectivity, regardless of the examiner. These assessments included evaluation of functionality such as general malfunctions, data loss, unrecoverable resets, and delayed effects such as pacing threshold changes, battery depletion, and signal interference. Documented CIED reports were analyzed from our local databank. In preparation for possible emergencies, cardiopulmonary resuscitation equipment was available at our facility at all times. Furthermore, an onsite resuscitation team and access to an intensive care unit were available. In our collective, an in-room audio and video system was used with each fraction of the RT. The authors had full access to and take full responsibility for the integrity of the data.

### 2.5. Statistical Analysis

Data collection for the presented project is conducted using an electronic database system, specifically Microsoft Excel from Microsoft GmbH in Unterschleißheim/Germany. Detailed descriptive statistics are provided for all data collected. For continuous data and scores, the mean, standard deviation, minimum, median, and maximum are calculated. Statistical analyses for mean and standard deviation (SD) were performed using the statistical software IBM SPSS software version 24.0.

## 3. Results

Sixteen patients, predominantly male (14) with a mean age of 77 (range: 56–85) years were enrolled in this study. The majority of patients suffered from esophageal cancer stage T3 (62.4%). Most patients suffered from cardiovascular risk factors, mainly hypertension (87.5%) and active or history of smoking (68.8%). The prevalence of coronary artery disease and HFrEF were high with 68.8% and 62.5%, respectively. Patient characteristics are shown in Table 1.

Among the cohort, 15 received treatment with IMRT and 1 patient was treated with 3D-CRT. Treatment characteristics are shown in Table 2. In 11 patients, the tumor was anatomically located in the lower part of the esophagus, in 3 patients between 20 and 28 cm from the row of teeth and in 1 patient in the upper part. The mean prescribed cumulative total dose to primary tumor and lymph node metastasis was 58.8 Gy (range: 36.0 Gy to 58.8 Gy). Information on radiation dose and fractionation, RT-technique, diagnosis, tumor site, type of CIED, manufacturer of the device, pacemaker-dependence, and information generated by the device was collected and is shown in Table 2 and Table 3. The mean distance between CIED and PTV was 10.0 cm (range 2.2–18.5 cm) (Figure 1). Patients with cumulative total dose greater than 3 Gy to CIED had a distance of 2.8 cm between PTV and CIED. These patients had a CIED from Medtronic.

Seven patients had ICD that received a mean total dose of around 2.2 Gy (range 0.96 Gy to 4.81 Gy). Additionally, 6 patients had PM that received a mean total dose of around 0.6 Gy (range: 0.2 Gy to 2.0 Gy), while 3 patients had CRT that received a mean total dose of around 2.9 Gy (range: 1.4 Gy to 3.0 Gy). The uncertainties on TPS dose values calculated for the PTV were range from 0.1% to 2.1% with the mean uncertainty of 0.4% (Table 2).

In the patient cohort, a total of 409 fractions were administered, with a corresponding 100% reporting rate for CIED assessments. There were no events such as set-ups or other changes of parameters in this study group. During the follow-up of 12 months, no case of sudden cardiac death was reported in the entire population. Detailed measurement results for the whole collective for pacing threshold, sensing threshold, and lead impedance of CIED before and after RT are shown in Table 4.

## 4. Discussion

The primary purpose of the present study focused on analyzing RT-related malfunctions with CIEDs in patients with esophageal cancer. Several studies showed that RT can result in both temporary and permanent damage to CIEDs [24,25,26,27]. In our study, we did not observe any severe CIED malfunction during or after RT of patients with esophageal cancer. However, comparing existing studies is difficult due to the lack of in vivo studies. Therefore, it is important to conduct more in vivo studies to better understand these malfunctions. Malfunctions such as the ‘runaway pacemaker’ can only be observed in vivo, where the adjusted stimulus thresholds of the CIEDs compete with the patient’s own rhythm. This cannot be imitated in in vitro experiments [28,29]. Previous findings indicate that adherence to RT guidelines for patients with CIEDs is important to avoid damaging the devices [17]. Our results, along with data from the literature, reveal that if CIED is near to the treatment field, it should be included into the planning computed tomography for correct estimation of the cumulative RT dose to the CIEDs. The CIED should not be part of the planning target volume (PTV) to minimize the dose at the CIED, as recommended by guidelines [16,30].

The effect of ionizing radiation can manifest itself in two different ways: first, proportional to the cumulative radiation dose, malfunctions arise due to the formation of electron–hole pairs; second, defects can emerge independently of the total dose and thus unpredictably. This stochastic event is called the single-event effect and appears to correlate with photon energy [31]. In our study, the mean cumulative total dose to ICD was 2.2 Gy (range 0.96 Gy to 4.81 Gy), 0.6 Gy (range 0.2 Gy to 2.0 Gy) to PM, and 2.9 Gy (range 1.4 Gy to 3.0 Gy) to CRT. The uncertainties in this study on TPS dose values calculated for the PTV were range from 0.1% to 2.1% with the mean uncertainty of 0.4%. This goes in line with the current literature. Several factors can influence CIED malfunction associated with RT. One critical factor is the maximum cumulative dose to the CIEDs. Cut-off values have been proposed since the 1980s, with a 2.0 Gy threshold mentioned in the first guidelines endorsed by the American Association of Physicists in Medicine in 1994 [4,32].

It is widely recognized that photon energy (≥15 megavolts (MV)) is a significant risk factor, with several malfunctions observed at energies ranging from 15 to 18 MV. In contrast, lower energies (<15 MV) seem to present a lower risk of resets or other types of malfunctions in CIEDs. This finding can be attributed to the increased rate of neutron production at higher energies, as neutrons can interfere with electronic components of devices, predominantly resulting in device resets. In our current study, we only used 6 MV photon energy for RT, thus the secondary neutron risk was very low based on several literature studies [4,24]. In synopsis of the available data and literature, the risk classification of patients should not be based exclusively on the location of the target volume but rather on patient-specific factors such as CIED dependence and history of previous episodes of ventricular tachycardia. Furthermore, the choice of photon energy should always be included in the risk classification. The largest existing multicenter study by Zaremba et al. demonstrated in a cohort of 560 patients who underwent RT with photons device malfunctions when using a median cumulative radiation dose of 46.5 Gy (ranges 20–70 Gy) and a median beam energy of 16.5 MV. There were no device malfunctions detected when using median cumulative radiation dose of 30 Gy (ranges 20–52 Gy) and 8 MV. The study revealed rate of malfunction is 2.5% for PMs and 6.8% for ICDs. The data indicate that the majority of observed errors were caused by electrical resets (78.6%). It is noteworthy that half of the radiation fields were found to be situated in the lower extremities or pelvis regions, exposing the CIEDs mainly to scattered radiation [33]. In our study, all patients were irradiated with a photon energy of 6 MV and mean cumulative total dose to ICD were 2.2 Gy, to PM 0.6 Gy, and to CRT 2.9 Gy. Therefore, our data and procedures align with current guidelines found in the literature. Our evaluation showed no device malfunctions, as all patients were treated with an energy of 6 MV.

Nevertheless, there are still no generally accepted recommendations from the manufacturers regarding the management of CIED patients during RT. This is partly due to the ongoing technological progress in these devices, and not all technical data are publicly available [4,16]. According to Boston Scientific and St. Jude Medical, no dose can be considered safe due to the potential effects of scattered radiation [34]. For example, Medtronic recommends a maximum dose of 5 Gy for PM, and the maximum dose for ICD ranges from 1–5 Gy, depending on the device model [17]. All manufacturers agree that direct irradiation of the device should be avoided. High-risk, secondary X-rays ≥10 MV are classified as they generate secondary neutrons which can result in clinical abnormalities [22,35,36]. Even though no adverse effects occurred in the present collective, nor in the long-term follow-up, it is important to consider certain factors before initiating RT. It is crucial to avoid irradiating the entire CIED directly and minimize the cumulative dose to the main body of the CIED [22].

To protect the CIED, some clinics use direct lead shielding. In our collective, shielding of the CIED was not necessary after consultation with our medical physics team and was not performed. There is a lack of clear and valid recommendations from the major manufacturers. For example, Medtronic recommends that the use of conventional X-ray shielding during RT does not adequately protect the CIED from neutron effects. Bourgouin et al. recommend the use of lead shielding based on the type of irradiation. For patients treated with IMRT, shielding is recommended only when using more than two anterior fields out of seven [37]. Before every treatment of patients, it is necessary to stratify the patients into risk groups and according to the planned radiation dose [21]. Low-risk patients are those who are not CIED dependent, and the pacemaker is not directly in the radiation field. Cumulative total dose of CIEDs in these low-risk patients is less than 2.0 Gy. Medium risk patients are those who are CIED dependent, and the pacemaker is not directly in the radiation field. Here, the cumulative total dose should be as well less than 2.0 Gy. High-risk patients are those who are CIED dependent and the CIED is not directly in the radiation field and the dose to the CIED is more than 2.0 Gy [16]. Tajstra et al. criticize the use of IMRT technique in CIED patients due to the assumed risk of increased scattered radiation exposure to the CIED. However, this assumption can be refuted as IMRT can be safely carried out even in CIEDs with target volumes only 2.5 cm away, and the scattered radiation exposure is far below any previously formulated photon radiation limits. IMRT enables the treatment of tumors in close proximity to the CIED by using conformal radiation fields. This technique limits photon energy to 6 MV in deep localized tumors, avoiding hotspots and the need for complex shielding measures or relocation of CIEDs. The relevance of investigating CIED problems within radiotherapy in conjunction with IMRT is increasing due to the eventual replacement of 3D-CRT with modern radiotherapy techniques such as IMRT in many oncological radiotherapy cases [38]. There are uncertainties in all technical measurements, including CIED readings. These are device and manufacturer specific. In the present study, the measurement uncertainties listed could be specified for the most frequently used CIED: stimulation impedances +/−10% with a resolution of 19 ohms and shock impedances +/−10% with a resolution of 2 ohms. In addition, a threshold +/−0.5 V relative to manual threshold for thresholds below 2.5 V and +/−1.0 V for higher thresholds. In our study, there are no major differences in the measured values before and after RT.

Due to the statistical and physical measurement uncertainty of the CIED devices and the small number of patients in the present study, the results demonstrated here should be interpreted with caution. Our available data indicate negligible alterations post-radiotherapy, which supports the claim that radiotherapy is a safe option for patients with esophageal cancer and CIED, provided that safety protocols are followed.

We acknowledge several limitations in our study. The retrospective nature of our study at a single center with a relatively small number of patients, limits the support for the conclusions regarding infrequent complications. Additionally, clinical reports and the presence of electromagnetic disturbance during or after RT could only be retrospectively reviewed. No serious adverse events in these elderly patient cohort with multimorbidity occurred during and after RT. The follow-up process may be biased by self-reporting as it involved direct telephone contact with patients. In the present study, continuous electrocardiographic monitoring was not performed during on table-time of irradiation, which may have missed possible malfunctions of the CIED during the single fraction. Certainly, the relevant measurement uncertainties of the CIED are also among the limitations existing in this study.

## 5. Conclusions

In our study, we did not observe any severe CIED malfunctions following each radiation fraction or after completion of RT in patients treated for esophageal cancer. Our findings reveal that electromagnetic disturbance is rare in patients with esophageal cancer during and after RT when there is a close collaboration between radiation oncologists, cardiologists, medical physicists, and radiation technologists. Strict selection of photon energy and alignment with manufacturer-recommended dose limits appear to be important. It is necessary to note that there are uncertainties in all technical measurements, including CIED readings which are device and manufacturer specific. In our study, there are no major differences in the measured values of the pacing threshold, sensing threshold, and lead impedance before and after RT. However, due to the statistical measurement uncertainty and the uncertainties associated with the physical measurement variables of the control devices, the small number of patients requires that the results be interpreted with caution. Based on the data obtained, risk of CIED dysfunction should be ideally determined for each patient prior to RT, and testing of CIED functionality should be performed to avoid life-threatening consequences. In addition, considering the work in the literature and the present study, the development of an up-to-date international guideline for RT treatment of patients with CIED appears to be desirable.

## Figures and Tables

**Figure 1 cancers-16-00555-f001:**
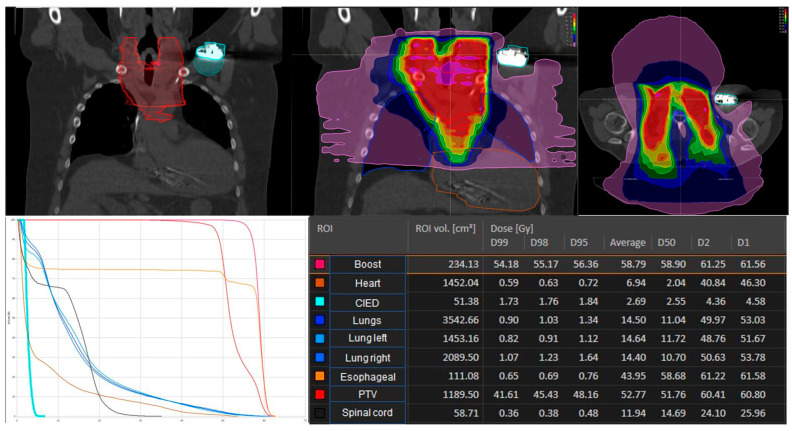
Radiation plan of a 62-year-old patient with esophageal cancer: distance and dose distribution to the cardiac implantable electronic device (CIED). The D95 in this patient was 1.84 Gy at the pacemaker. According to TPS, the uncertainty of the planned dose values for the PTV in this patient was 0.3%.

**Table 1 cancers-16-00555-t001:** Patient characteristics.

Characteristics	No of Patients (%)
Gender	
Male	14
female	2
Age at RT	
mean (range)	77 (56–85 years)
T-stage	
T1	1 (6.25%)
T2	2 (12.5%)
T3	10 (62.4%)
T4	3 (18.75%)
N-stage	
N0	2 (12.5%)
N+	14 (87.5%)
Cardiovascular risk factors	
Hypertension	14 (87.5%)
Hypercholesterolemia	8 (50.0%)
Diabetes mellitus	8 (50.0%)
Smoking	11 (68.8%)
Coronary artery disease	11 (68.8%)
Heart failure with reduced ejection fraction	10 (62.5%)
Prior percutaneous coronary interventions	5 (31.3%)
Prior aortocoronary bypass grafting	4 (25.0%)

**Table 2 cancers-16-00555-t002:** Treatment characteristics.

	*n* (%)
RT-technique	
3D-CRT	1 (6.25%)
IMRT	15 (93.75%)
Mean total dose PTV	58.8 Gy (range:36.0–58.8 Gy)
irradiation cervical lymph nodes	
yes	12 (75.0%)
No	4 (25.0%)
Mean PTV-volume	810 ccm (range 207–1539 ccm)
Mean total dose CIED	1.75 Gy (range 0.2–4.81 Gy)
Mean total dose ICD	2.2 Gy (range 0.96–4.81 Gy)
Mean total dose PM	0.6 Gy (range 0.2–2.0 Gy)
PTV	851 ccm (range 207–1539 ccm)
Uncertainties on planning dose values	0.4% (range 0.1–2.1%)

**Table 3 cancers-16-00555-t003:** Cardiac implantable electronic device (CIED) characteristics.

	*n* (%)
CIED type	
ICD	7 (43.8%)
PM	6 (37.5%)
CRT	3 (18.8%)
PM dependence	
Dependent (</= 30 bpm)	2 (22.2%)
Not dependent (>30 bpm)	7 (77.8%)
Missing	0 (0%)
Manufacture	
Medtronic	10 (62.5%)
St. Jude Medical	5 (31.25%)
Boston Scientific	1 (6.25%)

**Table 4 cancers-16-00555-t004:** Measurement results of cardiac electronic devices before and after RT.

Patients	Pacing Threshold (Volt)		Sensing Threshold (mV)		Lead-Impedance (Ohm)	
	before RT	after RT	before RT	after RT	before RT	after RT
1	0.6	0.5	5.8	5.8	388	390
2	1.0	1.12	5.3	5.3	300	292
3	0.75	0.75	20.2	21.4	478	475
4	3	3	2.3	4.2	544	528
5	0.75	0.5	9.6	7.6	350	363
6	0.75	0.87	22.4	22.4	671	689
7	0.75	0.75	6	5.1	456	456
8	1.0	1.25	6.8	8.0	700	779
9	1.0	1.0	11	19.9	480	760
10	0.75	0.75	11.3	12.1	475	475
11	0.75	0.75	6.8	6.8	719	674
12	0.5	0.5	7.2	7.3	475	463
13	0.5	0.5	22.4	22.4	848	756
14	0.9	0.8	7.2	11.4	428	459
15	1.0	1.0	6.3	11.8	375	400
16	0.75	0.75	5.8	5.8	380	399
Mean/± SD	0.9 ± 0.6	0.9 ± 0.6	9.8 ± 6.1	11.1 ± 6.5	504.2 ± 148.8	522.4 ± 152.1

## Data Availability

The authors confirm that the data supporting the findings of this study are available within the article. Due to the nature of this research, participants of this study did not agree for their data to be shared publicly, so supporting data is not available.

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
