# Peer review of "Effects of Ionizing Radiation on Cardiac Implantable Electronic Devices (CIEDs) in Patients with Esophageal Cancer Undergoing Radiotherapy: A Pilot Study"

_cancers, 2024, doi:10.3390/cancers16030555_

Round 1

Reviewer 1 Report

Comments and Suggestions for Authors

Uzun and colleagues conducted an analysis of radiotherapy-related malfunctions of CIEDs. The overall discussion is deemed sufficient, and minor revision is recommended.

1. In Figure 1, it would be beneficial to discuss the rationale for selecting the patient. Consideration of factors like cardiovascular risk and tumor/node stage could provide additional context.

2. The format of figures and tables needs careful revision, addressing issues such as the red wavy line in Figure 1 and abnormal alignment in Table 4 for improved clarity.

Author Response

Thank for your constructive suggestions. Figure 4 and Tables were adapted like your suggestion and red wavy line deleted. In Material and methods part page 3 line 122 we completed the sentence into the following and highligted red:

In the entire study group, analyses were performed to evaluate the distance of the RT plan and the dose at the CIED (D95). An example of a patient with esophageal carcinoma located very close to the CIED and cervical lymph node metastases is shown in Figure 1 to illustrate that in radiation planning, distances from the tumor and CIED are important for dosimetric planning.

Reviewer 2 Report

Comments and Suggestions for Authors

The study aims could be of some interest however the very small study population is a major limitation of the study 

Comments on the Quality of English Language

English is fine 

Author Response

Thank you to your suggestions, we improved the whole manuscript.

We agree with your comments, that the number of patients in our study is low. Due to the low incidence of esophageal cancer in Germany and the resulting scarcity of patients with CIEDs, even in one of the largest European cancer centers (University Hospital Heidelberg/Germany) cannot obtain a larger number of cases during this extended study period. Our study aims to provide a foundation for potential future multicenter analyses of radiotherapy and CIED dysfunction in esophageal cancer. The literature currently underrepresents this topic.

Reviewer 3 Report

Comments and Suggestions for Authors

The manuscript discusses the effects of radiotherapy (RT)-related malfunctions on cardiac implantable electronic devices (CIEDs) in patients with esophageal cancer. The study found no severe CIED malfunctions during or after RT, indicating adherence to RT guidelines for patients with CIEDs is critical to prevent damage. The results suggest that the CIED should be included in planning computed tomography for accurate dose estimation and should not be part of the planning target volume to minimize dose exposure.

It highlights the importance of avoiding high photon energy (≥15 megavolts) due to the increased risk of malfunctions, with lower energies (<15MV) presenting a lower risk. The study also notes that patient-specific factors, such as CIED dependence and previous episodes of ventricular tachycardia, should be considered alongside the location of the target volume for risk classification.

The authors acknowledge the limitations of their study, including its retrospective nature, the small number of patients, and the potential biases in follow-up processes. They also mention the absence of continuous electrocardiographic monitoring during RT, which could miss possible CIED malfunctions. The study concludes that RT appears to be a safe option for patients with esophageal cancer and CIED, provided safety protocols are followed. It suggests that a collaborative approach between various medical professionals is beneficial and emphasizes the need for individual risk assessment of CIED dysfunction before RT. The development of updated international guidelines for RT treatment of patients with CIED is proposed as desirable.

For improvements, the authors might consider expanding the patient cohort in future studies for more robust data, implementing real-time monitoring of CIED function during RT, and collaborating with device manufacturers for updated guidelines reflective of technological advances in CIEDs.

I only have some suggestions:

-please state the aim of the paper in the last paragraph of the Introduction

-please restructure materials and methods into subtitles, it will be easier for the readership

-baseline characteristics table should be revised..

Comments on the Quality of English Language

minor

Author Response

Thank you to your suggestions. Baseline Table was improved. We also  improved the introduction part to :

The objective of this study is to investigate RT-related dysfunctions of CIEDs in patients with oesophageal cancer.

We also included subtitles in material an methods part:

2.1 Evaluation

This retrospective study was performed following institutional guidelines and the Declaration of Helsinki of 1975 in its most recent version. Ethical approval for the study was given from the local ethics committee at University Hospital Heidelberg (S-123/2021). Clinical, operative, and hospital course records were reviewed. Local databank was screened for patients with CIED, esophageal cancer and RT between 2012 to 2022 at the University Hospital Heidelberg. RT was performed at the Department of Radiation Oncology at University Hospital Heidelberg using either intensity-modulated radiotherapy (IMRT) or three-dimensional radiotherapy (3D-CRT).

2.2 CIED details

Prior to RT, all patients underwent systemic evaluation of the CIED functionality at the Department of Cardiac Electrophysiology of the Heidelberg University Hospital. Regarding CIED information, data collection at baseline (pre-RT) was performed including for different types of CIEDs (PM, ICD, CRT). Also, device location, manufacturer, number of leads, primary indication for implantation, PM dependency (defined as intrinsic rhythm ≤ 30 bpm), and history of ventricular arrhythmias (for ICD) were recorded. Naturally, as with any technical measurement, there are also uncertainties in the CIED measurements. There are no universally valid measurement uncertainties available here, rather it depends on the individual CIED and its measuring devices. After consultation with the manufacturers, we were able to provide the following measurement uncertainties for the most common CIED used in our study: Pacing impedances +/-10% with resolution of 19 ohm, Shock impedances +/-10% with resolution of 2 ohms. Threshold +/-0.5V relative to manual threshold for thresholds less than 2.5V and +/-1.0V for higher thresholds.

2.3 Treatment planning and treatment characteristics

During RT-treatment planning process, all patients underwent CT simulation. RT was administered using 6 MV photons after treatment planning process, no neutron damage was possible at these energy levels. There was no patient treated with electrons. Clinical target volume definition was based on contrast-enhanced CT scans, included the primary tumor region as well as nodal involvement according to the International Commission on Radiation Units and Measurements (ICRU) definition for esophageal cancer (Wu et al., 2015). A planning target volume (PTV) margin of 5 to 8 mm was added depending on the applied technique. In the planning CT the CIED was encompassed for all patients completely. In planning process all CIEDs were restricted to total doses of 3.0Gy or less and were not located within the treatment field. Relocation of the device is generally considered when this precondition is not possible, which was not the case in our study cohort. In the entire studied group, analyses were made to assess the distance of the RT plan and the dose at the CIED (D95). An example is shown in figure 1.There was no need to use asynchronous stimulation of pacemakers or deactivation of antitachycardia treatment of ICD's through reprogramming or magnet placement. We did not perform continuous monitoring by electrocardiography.

2.4 CIED follow up

All patients were seen by a specialist at the department of Cardiac Electrophysiology of the Heidelberg University Hospital after every single fraction and 6 weeks as well as 12 months after RT. These assessments included evaluation of functionality such as general malfunctions, data loss, unrecoverable resets, and delayed effects such as pacing threshold changes, battery depletion and signal interference. Documented CIED reports were analyzed from our local databank. In preparation for possible emergencies, cardiopulmonary resuscitation equipment was available at our facility at all times. Furthermore, an onsite resuscitation team and access to an intensive care unit were available. In our collective, an in-room audio and video system was used with each fraction of the RT. The authors had full access to and take full responsibility for the integrity of the data.

2.5 Statistical Analysis

Statistical analyses for mean and standard deviation (SD) were performed using the statistical software IBM SPSS software version 24.0 (table 4).

Round 2

Reviewer 2 Report

Comments and Suggestions for Authors

Dear Authors 

Due to the limited number of patients included in the study, I suggest changing the title "Effects of ionizing radiation on cardiac implantable electronic devices (CIEDs) in patients with esophageal cancer undergoing radiotherapy: A pilot study"

Author Response

Dear reviewer,

Thank you for your constructive criticism. We have changed the title of our work as you requested. We appreciate your feedback.

Sincerely,
DD Uzun 

Reviewer 3 Report

Comments and Suggestions for Authors

The authors have addressed all my comments. Thank you.

Author Response

Dear Reviewer,

Thank you for your feedback. We have incorporated all necessary points and are pleased to have done so. We are also happy to improve our manuscript.